# Phase Lag Entropy as a Surrogate Measurement of Hypnotic Depth during Sevoflurane Anesthesia

**DOI:** 10.3390/medicina57101034

**Published:** 2021-09-28

**Authors:** Kyung-Mi Kim, Ki-Hwa Lee, Jae-Hong Park

**Affiliations:** 1Department of Anesthesiology and Pain Medicine, Asan Medical Center, University of Ulsan College of Medicine, Seoul 05505, Korea; sumsonyo@gmail.com; 2Department of Anesthesiology and Pain Medicine, Haeundae Paik Hospital, Inje University School of Medicine, Busan 47392, Korea; H00150@paik.ac.kr

**Keywords:** anesthesia, electroencephalogram, pharmacodynamics, phase lag entropy, sevoflurane

## Abstract

*Background and Objectives*: Phase lag entropy, an electroencephalographic monitor, evaluates the variety in temporal patterns of phase relationship between frontal and prefrontal brain region. Phase lag entropy can reflect the depth of anesthesia induced by propofol, but the association between sevoflurane and phase lag entropy has not been elucidated. This study examined the effect of sevoflurane on phase lag entropy during induction of general anesthesia. We also explored the pharmacodynamic model between end-tidal anesthetic concentration and electroencephalographic monitor. *Materials and Methods*: A total of 20 patients were enrolled. General anesthesia was produced by escalating the sevoflurane (1 vol% up to 8 vol%). The relationship between phase lag entropy and end-tidal anesthetic concentration was analyzed. A non-linear mixed-effects model was used to get the relationship of pharmacodynamics between the end-tidal sevoflurane concentration and phase lag entropy. Mean blood pressure, heart rate, and the modified observer’s assessment of alertness/sedation scale were also recorded during sevoflurane anesthesia. *Results*: As level of sedation increased, phase lag entropy decreased. A significant correlation was showed between phase lag entropy and end-tidal sevoflurane concentration (r = −0.759, *p* < 0.001). The correlation coefficient between the modified observer’s assessment of alertness/sedation scale and phase lag entropy was 0.731 (*p* < 0.001). The pharmacodynamic factors assessed by the sigmoid *E_max_* model were *E*_0_ = 84.9, *E_max_* = 42, *C_e_*_50_ = 1.81, γ = 4.78, and *k_e_*_0_ = 0.692. The prediction probability of phase-lag entropy for measuring the modified observer’s assessment of alertness/sedation scale and end-tidal sevoflurane concentration were 0.764 and 0.789, respectively. With the increasing concentration of sevoflurane, mean blood pressure decreased, but heart rate did not change. *Conclusions*: The continuing escalation in end-tidal sevoflurane concentration caused a decline in phase lag entropy. Phase lag entropy can serve as an indicator of hypnotic depth in patients receiving sevoflurane anesthesia.

## 1. Introduction

Anesthetic agents cause unconsciousness and inhibit the feedback connectivity, which is characteristic of the conscious state [1]. Anesthetics block the capability of the brain to incorporate information, which leads to simplicity of communication among brain territories [2,3].

Alterations in functional connectivity and interruptions in frontal electroencephalographic (EEG) communication have been reported during general anesthesia [1,4]. Modern theories of consciousness state that the brain exhibits particular properties during the conscious state, but these patterns are absent in the unconscious state [5]. Most EEG monitors used to check depth of anesthesia depend on the temporal features of single-channel EEG. Single-channel EEG monitors may not present spatial or functional connectivity information of the brain. Therefore, depth of anesthesia monitors may be more appropriate if they reproduce functional connectivity of dissimilar brain territories [6].

Directional feedback connectivity from anterior to posterior brain regions is inhibited by propofol and sevoflurane [1]. Similar neural circuit mechanisms are involved, as evident by the similar EEG dynamics induced by sevoflurane and propofol [4]. Looking for common EEG features, propofol and sevoflurane anesthesia were depicted by alpha (8–12 Hz) and slow (<1 Hz) oscillations [4]. The EEG features of sevoflurane were discriminable from that of propofol because of increased power and coherence in the theta wave (4–7 Hz) [4].

The phase lag entropy monitoring device (PLEM100) is a new four-channel EEG monitoring device that has been recently developed. Phase lag entropy exhibits the depth of anesthesia by computing the various interconnection of phase relationships between two EEG signals [7]. Recently developed phase lag entropy has been closely associated with the level of propofol sedation [8,9,10,11,12]. No study has evaluated the relationship between phase lag entropy and sevoflurane anesthesia to the knowledge of the authors.

The primary purpose of this study was to evaluate the relationship between phase lag entropy and sevoflurane concentration. We attempted to obtain a population pharmacodynamic model for sevoflurane concentration and phase lag entropy during the induction phase.

## 2. Materials and Methods

### 2.1. Patient Population

This study was accepted by the Institutional Review Board of Haeundae Paik hospital (Approval number: 2017-07-636), and was registered in a clinical trial registry (http://cris.nih.go.kr; accessed on 27 December 2017) with the registration number KCT0003180. Informed consents were gained from all the patients. American Society of Anesthesiologists physical status Class I to II, adult patients (>20 years old) who were scheduled for elective surgery were included. Exclusion criteria were obesity (body mass index >30 kg/m^2^), and preoperative abnormal cardiopulmonary, renal, and hepatic function. Patients who had an abuse history of drug or alcohol were also excluded.

### 2.2. Study Procedure

Patients did not receive any premedication before surgery after fasting overnight. Electrocardiography, non-invasive blood pressure, pulse oximetry, and end-tidal carbon dioxide concentration measurements were applied to all the patients. Frontal raw EEG signals were recorded using the PLEM sensor. PLEM sensors were located at FP1 (L1), FP2 (R1), AF5 (L2), and AF6 (R2). The ground electrode was at Fpz, and the reference electrode was at position T3 on the temporal area of the face. General anesthesia was induced by escalating the sevoflurane (1 vol% to 8 vol%) with 100% oxygen using mask ventilation. When the end-tidal concentration of sevoflurane did not show any further change, we increased sevoflurane concentration by 1 vol% [13]. If the patient could not ventilate spontaneously during mask ventilation, intravenous (IV) rocuronium (0.6 mg/kg) was administered to the patients for muscle paralysis. Normocarbia (35–45 mmHg) was maintained because hyperventilation may induce the presence of epileptiform patterns in the EEG, which produces erroneous values in depth of anesthesia monitors [14]. EEG measurements were completed when the end-tidal sevoflurane concentration did not show any change after 8 vol%. We measured the end-tidal concentration of sevoflurane using a Datex-Ohmeda gas analyzer. The modified observer’s assessment alertness/sedation scale was used to assess the level of consciousness; 5 = responds readily to name spoken in normal tone, 4 = lethargic response to name spoken in normal tone, 3 = responds only after name is called loudly of repeatedly, 2 = responds only after mild prodding or shaking, 1 = does not respond to mild prodding or shaking, 0 = does not respond to noxious stimulus. [15]. Loss of responsiveness to verbal or tactile stimulus was defined as a condition in which a patient did not respond to the verbal command and loss of eyelash reflex (modified observer’s assessment alertness/sedation scale score = 2).

### 2.3. Correlation between Phase-Lag Entropy and End-Tidal Sevoflurane Concentration

The phase lag entropy value was calculated using the method suggested by Lee et al. [7]. We revealed the end-tidal sevoflurane concentration and phase-lag entropy values for calculating the Pearson’s correlation coefficients. Pearson’s correlation coefficient is scaled such that it ranges from −1 to +1. There is no linear association when coefficient is 0, and the relationship gets stronger and ultimately approaches a straight line.

### 2.4. Pharmacodynamic Modeling

For pharmacodynamic modeling using the non-linear mixed-effects modelling software NONMEM VII Level 5 (ICON Development Solutions, Dublin, Ireland), 690 points of phase-lag entropy values were selected. Convolution which is based on the ‘connect the dot’ approach can calculate the effect-site concentrations as time passes [16]. A sigmoid *E_max_* model was used to analyze the relation between end-tidal anesthetic concentration and values of phase lag entropy.

The inter-individual random variability was estimated using a log-normal distribution:$$P_{i}=P_{TV}e^{\eta_{i}}$$

The comparison of alternative structural models was based on a likelihood ratio test with NONMEM’s objective function value (OFV). A change in OFV between models of >3.84 was deemed statistically significant with a *p* value < 0.05.

A nonparametric bootstrap analysis was done using the internal model validation fit4NM 4.6.0 (http://www.fit4nm.org/download/246, accessed on 3 November 2020). Random sampling generated from original data set produced a total of 2000 bootstrap resamplings. The median values and the 2.5–97.5 percentiles of the non-parametric bootstrap replicates of the final model were compared with the final pharmacodynamic model parameter estimates.

### 2.5. Statistical Analysis

The data are presented as number and percentage for categorical variables, mean ± SD for normally distributed continuous variables, and median (interquartile range) for non-normally distributed continuous variables. The differences in characteristics of study participants according to assessment time were analyzed using the generalized linear mixed model with Bonferroni’s post-hoc test. The receiver operating characteristic curve analysis was performed to assess the sensitivity and specificity of phase-lag entropy for predicting the modified observer’s assessment alertness/sedation scale score ≤ 2. For graphical visualization, the error bar chart and scatter plot are displayed.

Statistical analyses were performed using MedCalc 19.8 (MedCalc software Ltd., Ostend, Belgium) and SPSS 24.0 (SPSS Inc., Chicago, IL, USA) for Windows (Microsoft Corporation, Redmond, WA, USA). *p* values less than 0.05 were considered statistically significant. Prediction probability was used to assess the correlation of end-tidal sevoflurane concentrations with phase lag entropy values and of each patient’s consciousness level measured using the modified observer’s assessment alertness/sedation scale score. Prediction probability values were calculated using the Somer’s d cross-tabulation statistic in fit4NM 4.6.0 [17].

## 3. Results

A total of 20 patients were enrolled. Of the 20 patients, patients were excluded because of loss of phase-lag entropy data (*n* = 2), loss of sevoflurane data (*n* = 4), and inclusion violation (*n* = 1). The demographic features are exhibited in Table 1.

The baseline value of phase lag entropy before sevoflurane administration was 85.23 ± 23.19. As end-tidal sevoflurane concentration gradually increased, the phase lag entropy and mean blood pressure tended to decrease significantly (Figure 1), while the heart rate did not change.

The confidence intervals (CI) of R are −0.8651 to −0.7811 for PLE (A) and −0.6186 to −0.4243 for mean blood pressure (B), respectively. The whiskers extend to the 10th and 90th percentile values. More extreme values (circle) are plotted individually (C). Data are expressed as mean ± SD (D). PLE: phase lag entropy. MOAA/S scale: modified observer’s assessment of alertness/sedation scale.

The pearson correlation coefficients between end-tidal sevoflurane concentration and phase lag entropy and between modified observer’s assessment alertness/sedation scale and phase-lag entropy were −0.759 (*p* < 0.001) and 0.731 (*p* < 0.001), respectively. The prediction probability of phase lag entropy for measured modified observer’s assessment alertness/sedation scale were 0.764 (SD (95% CI); 0.037 (0.692–0.837)) and for end-tidal sevoflurane concentrations were 0.789 (0.007 (0.775–0.804)).

The final pharmacodynamic parameters assessed using the sigmoid *E_max_* model were *E*_0_ = 84.9, *E_max_* = 42, *C_e_*_50_ = 1.81, γ = 4.78, and *k_e_*_0_ = 0.692. As shown in Table 2, the relative standard error for each fixed-effect parameter was <50%, indicating acceptable precision of the parameter estimates. In addition, the median parameter estimated resulting from the nonparametric bootstrap were reasonably close to the respective parameter estimates from the final model, representing good stability and reliability of the final pharmacodynamic model. The goodness-of-fit plots for the final model showed adequate model performance (Figure 2). The relationship between the measured vs. predicted concentrations illustrated that the predictions were unbiased and appropriate for the population and the individual subjects in this study. The plot of phase lag entropy over time for each participant as described by the model of pharmacodynamic are illustrated in Figure 3.

As shown in Figure 2, plots of weighted residual (WRES) against population-predicted or individual-predicted concentrations dispersed around zero, representing no systemic bias.

The area under the curve and receiver operating characteristic curve of phase lag entropy, showing the ability of the depth of anesthesia monitor to discriminate between modified observer’s assessment alertness/sedation scale ≤ 2 and modified observer’s assessment alertness/sedation scale score > 2, are presented in Figure 4. A phase lag entropy ≥ 75 identified modified observer’s assessment alertness/sedation scale score ≤ 2 with a sensitivity of 86.2% and a specificity of 95.0%, indicating that phase lag entropy adequately reflects hypnotic depth.

## 4. Discussion

This is the first study to evaluate the relationship between concentration of sevoflurane and phase lag entropy. The continuing escalation in end-tidal sevoflurane concentration caused a decline in phase lag entropy during induction of sevoflurane anesthesia. We detected a relation in close arrangement with the sigmoid *E_max_* model and deduced that phase lag entropy has potential as an indicator of hypnotic depth during sevoflurane anesthesia.

Variations of EEG depending on increasing concentration of sevoflurane were studied with state and response entropy as well as with BIS [18,19]. Therefore, knowledge about the effect of gradual increase in sevoflurane concentration on phase lag entropy was required. When exploring the relationship among drug dose-concentration-effect, the study of population pharmacodynamics can serve as a useful tool. The common pharmacodynamic models that apply in the field of anesthesiology are effect-compartmental, turnover, drug-receptor binding, and drug-interaction models [20]. The effect-compartmental model was formalized and first applied using response measurements obtained during and after administration of d-tubocurarine [21]. It is widely used for explaining drug responses. The sigmoid *E_max_* model adequately expressed the relationship between sevoflurane and phase-lag entropy effect in our study. Compared with another similar study [13], gamma, representing the steepness of the concentration-response curve, was higher in our study (1.27 vs. 4.78).

Several processed EEG monitors have been applied for the quantification of sevoflurane anesthesia effects [13,22]. Each EEG monitors have their own algorithms to calculate the index for the target of general anesthesia [23]. For example, 40–60 of BIS, 25–50 of PSI, and 40–60 of PLE is a range of index for adequate general anesthesia [11,23]. Phase lag entropy also showed high prediction probability in our study. An indicator of level of consciousness which predicts depth of anesthesia will reveal a prediction probability value of 1, while an indicator that conducts on chance (50:50) will exhibit a prediction probability value of 0.5 [17]. Prediction probability is the probability of an event that is calculated from obtainable data. Soehle et al. [24] discerned a high level of prediction probability of BIS (0.80 ± 0.11) and PSI (0.79 ± 0.09) during sevoflurane anesthesia in another studies.

Phase lag entropy is a new monitor that has recently been certified in South Korea. Although there are no studies on the pharmacodynamic effects of sevoflurane, there are several studies related to propofol sedation. Sevoflurane acts on glycine receptor and GABA_A_, activating potassium channels, and inhibiting excitatory neurotransmitter receptors [1,2]. Non-irritant and pleasant-smelling characteristics of sevoflurane render it more acceptable for the induction of anesthesia [25]. However, the ionic mechanism and target receptors of propofol is different from sevoflurane. Since each anesthetic agent has distinct characteristics in the EEG [26], examining the ability of EEG monitor by drugs would be more helpful to anesthesiologists. Ki et al. [9] showed the correlation coefficients (R = −0.621, *p* < 0.001) and prediction probability (0.646) between phase lag entropy and effect-site concentration of propofol. Jun et al. [8] demonstrated that phase-lag entropy was closely correlated with the modified observer’s assessment alertness/sedation scale during propofol sedation (r = 0.755, *p* < 0.001) and prediction probability value (0.731). Corresponding with this result, our study also showed a high correlation between end-tidal sevoflurane concentration/sedation score and phase-lag entropy. With a deepening sedation level, the phase lag entropy values decreased from 87 (median) at a modified observer’s assessment alertness/sedation scale of 5 to 45 at a modified observer’s assessment alertness/sedation scale of 0.

There has been a lot of interest in the consciousness. The reduction in the dynamic repertoire of brain states, observed by Hudetz et al. [27], supports the information integration theory of consciousness. The status of consciousness is more closely connected with the temporal dynamics of the functional network configuration between brain areas [27,28]. Lack of diversity of phase lag pattern is linked with breaking of consciousness, and phase lag entropy quantifies this diversity of phase lag patterns. The change is most stated in frontal and prefrontal montage connections [7]. Therefore, diversity of connectivity configurations, measured by PLE, might be a more precise anesthetic depth indicator.

There are several limitations of this study. First, plotting phase lag entropy versus end-tidal sevoflurane concentration did not reveal hysteresis. This might be because we simply evaluated end-tidal sevoflurane concentration in the rising loop and did not consider the descending loop. We evaluated patients undergoing elective surgery, and this restricted the design of our study. A duration of 40 min or more is required for observing hysteresis [29]. However, such a long period during induction is difficult to acquire in the clinical field, even if the surgeon is very generous. Second, sevoflurane was administered via a facemask instead of an endotracheal tube. If we applied endotracheal tube, we could exclude the possibility of sevoflurane leakage. Third, most studies examining the relationship between phase-lag entropy and propofol employed pairwise comparisons with BIS [9,10,12]. However, we did not use pairwise comparisons with BIS. This should be the focus of further follow-up studies. Fourth, time delay between the detection of EEG patterns and their transformation into a digital signal may impair the actual depth of anesthesia, which must be taken into consideration, and has not yet been studied with phase lag entropy [30]. Fifth, the presence of epileptiform patterns in patient EEG were found to abnormally increase or decrease the state entropy values, and the influence on their possible in the present study on the phase lag entropy value is unknown and requires similar further studies [19].

## 5. Conclusions

Phase lag entropy, a new method for EEG monitoring, can serve as a useful indicator of hypnotic depth in patients receiving sevoflurane. Further evaluations of the clinical applications of various dose of sevoflurane and other sedatives are required to confirm the consistency of phase lag entropy.

## Figures and Tables

**Figure 1 medicina-57-01034-f001:**
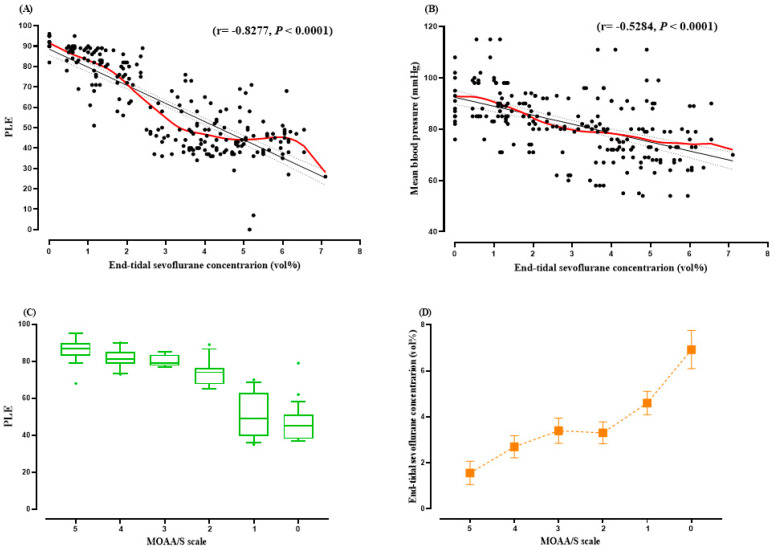
(**A**) Correlation between phase lag entropy, (**B**) mean blood pressure, and end-tidal sevoflurane concentration during sevoflurane anesthesia. (**C**) Plot of the modified observer’s assessment of alertness/sedation scale and phase lag entropy. (**D**) End-tidal sevoflurane concentration corresponding to each modified observer’s assessment of alertness/sedation scale.

**Figure 2 medicina-57-01034-f002:**
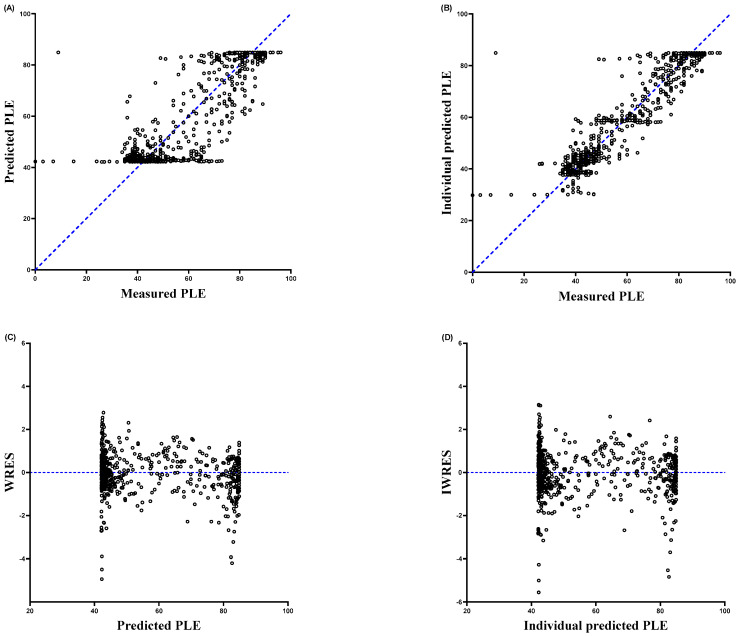
Weighted residual was calculated using the following equation: Weighted residual = (measured − predicted)/predicted. (**A**) predicted vs. observed phase lag entropy, (**B**) individual predicted vs. observed phase lag entropy, (**C**) weighted residuals (WRES) vs. predicted phase lag entropy, (**D**) individually weighted residuals (IWRES) vs. predicted phase lag entropy.

**Figure 3 medicina-57-01034-f003:**
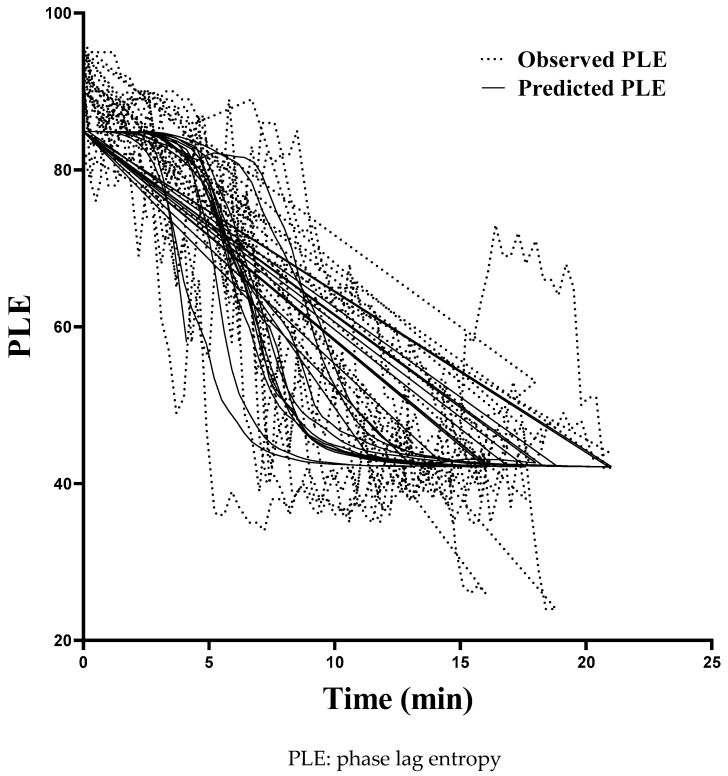
The phase lag entropy-time relationship for each participant as described by the model of pharmacodynamic.

**Figure 4 medicina-57-01034-f004:**
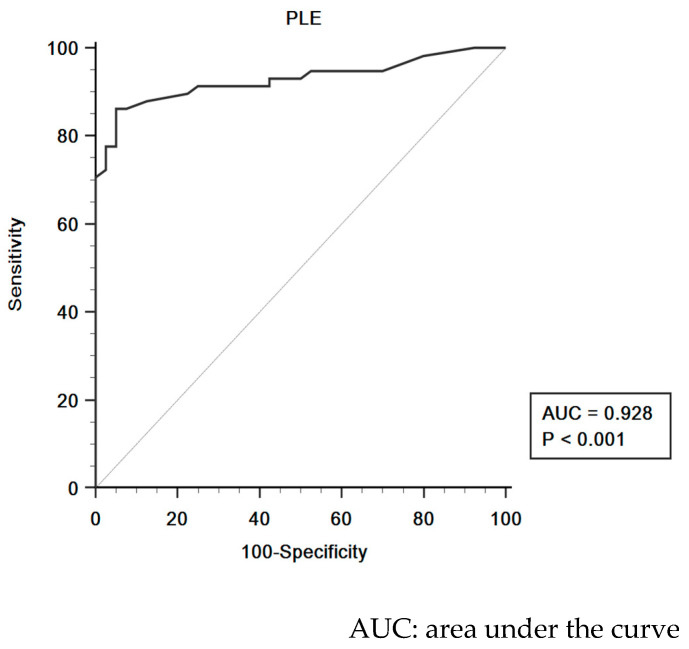
Receiver operating characteristic curves of phase lag entropy.

**Table 1 medicina-57-01034-t001:** Demographic features of the subjects.

	Variables
Age (years)	50.00 (37.00–53.25)
Sex (male/female)	8/5 (62%/38%)
Height (cm)	166.50 (160.80–171.28)
Weight (kg)	66.10 (62.75–73.83)
ASA classification (Ⅰ/Ⅱ)	8/5 (62%/38%)
BMI (kg/m^2^)	25.51 (23.46–26.24)
Anesthesia time (min)	185.00 (116.25–202.50)
Type of surgery	
Neck mass removal	3 (23%)
Tympanoplasty	8 (62%)
Parotid tumor removal	2 (15%)

Values are presented as counts (percentages) or median (interquartile range) deviation. ASA: American society of anesthesiologists, BMI: body mass index.

**Table 2 medicina-57-01034-t002:** The estimates pharmacodynamic parameters and median values (2.5–97.5%) of the final pharmacodynamic model.

Parameter	Estimates (RSE, %)	CV (%)	Median (2.5–97.5%)
*E* _0_	84.9 (0.79)	‒	85.0 (82.9–86.8)
*E_max_*	42 (5.76)	18.73	41.6 (38.7–45)
*C_e_*_50_,vol%	1.81 (8.95)	27.82	1.80 (1.8–2.1)
γ	4.78 (10.61)	‒	4.92 (4.5–5.5)
*k_e_* _0_	0.692 (33.82)	67.68	0.673 (0.42–0.80)
σ _2_	1	‒	‒

CV: coefficient of variation, RSE: relative standard error.

## Data Availability

The datasets are available from the corresponding author on reasonable request.

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
