# Peer review of "Phase Lag Entropy as a Surrogate Measurement of Hypnotic Depth during Sevoflurane Anesthesia"

_medicina, 2021, doi:10.3390/medicina57101034_

Round 1

Reviewer 1 Report

This paper is useful since phase lag entropy can serve as an indicator
of hypnotic depth in patients receiving sevoflurane anesthesia.

Reviewer 2 Report

Kim et al. investigated the correlation between phase lag entropy (PLE) – a relatively novel EEG parameter – and sevoflurane concentration. In addition, they evaluated the behavioral changes concomitant with the increasing anesthetic gas concentration. They found a strong correlation between PLE, MOAA/S and sevoflurane concentration. So the PLE might be an additional parameter which can help the attending anesthesiologist to assess patient’s consciousness. However, I am not sure that it is outperforming established processed EEG parameters, e.g. PSI, BIS, Narcotrend etc. So the clinical relevance of PLE still has to be proven. Nevertheless, this study with some minor flaws adds new aspects to the present knowledge. Please find my suggestions for changes below.

Introduction:

  • First Paragraph: During GA you can usually observe alpha activity in the frontal cortex. So in my opinion, the phrase “Unconsciousness caused by anesthetics usually happen together with the cessation of activity in the cerebral cortex.” is wrong.
  • Please specify the theta band range.

Methods:

  • Can you describe the placement of the electrodes in regard to the 10-20 EEG system?
  • The term unconsciousness is not clearly defined. You might want to use loss of responsiveness to verbal or tactile stimulus.
  • Please explain Pearson’s correlation coefficient.
  • Please explain MOAA/S or show a table in the supplement.
  • How many patients received rocuronium? Did you evaluate if rocuronium had an effect on PLE?

Results:

  • Table 1: Since n=13, I would prefer median (1st and 4th quartile)
  • Please add type of surgeries to table 1
  • Please add the term for the regression models.
  • Please add to Figure 1 the scatter plot for PLE vs. MOAA/S and end tidal sevoflurane concentration vs. MOAA/S.
  • Figure 4 is missing

Discussion:

  • BIS and PSI are not pharmacodynamic parameters. They are processed EEG parameters calculated by different algorithms. The cited studies did not directly compare PSI during propofol vs. PSI during sevoflurane anesthesia. So in my opinion they do not support the authors assumption.
  • Page 9, second paragraph: “For example, 40-60 of BIS,…” Please add citations.

Author Response

We would like to thank you for taking the time to review our article. We have made some corrections and clarification in the manuscript after going over the reviewers' comment. Please see the attachment. We hope that the revised manuscript will better meet the requirements of your journal for publication. 

Reviewer 3 Report

Dear authors,

Well done a study!!!

Whereas some details mus be attended.

In the methodology section some more details concerning the technique of volatile induction general anaesthesia are required - see simillar article or write that you used simillar technique, provide the citation, what closes the discussion.

Stasiowski MJ, Marciniak R, Duława A, Krawczyk L, Jałowiecki P. Epileptiform EEG patterns during different techniques of induction of general anaesthesia with sevoflurane and propofol: a randomised trial. Anaesthesiol Intensive Ther. 2019;51(1):21-34. doi: 10.5603/AIT.a2019.0003. Epub 2019 Feb 6. PMID: 30723886.

Yoy must definitely define when was every patient paralysed with rocuronium and if data was collected afterwards. This is crucial because the use of rocurominum was proven to influence the state entropy values. See my comment in the limitations section. Schuller PJ, Newell S, Strickland PA, Barry JJ. Response of bispectral index to neuromuscular block in awake volunteers. Br J Anaesth. 2015 Jul; 115 :i95-i103. doi:10.1093/bja/aev072

in the discussion section you write: 

Therefore, knowledge about the 
effect of gradual increase in sevoflurane concentration on phase lag 
entropy was required. I would add: likewise it was studied with state and response entropy as well as with BIS. 

Stasiowski MJ, Duława A, Król S, Marciniak R, Kaspera W, Niewiadomska E, Krawczyk L, Ładziński P, Grabarek BO, Jałowiecki P. Polyspikes and Rhythmic Polyspikes During Volatile Induction of General Anesthesia With Sevoflurane Result in Bispectral Index Variations. Clin EEG Neurosci. 2020 Nov 26:1550059420974571. doi: 10.1177/1550059420974571. Epub ahead of print. PMID: 33241952.

Stasiowski M, Duława A, Szumera I, Marciniak R, Niewiadomska E, Kaspera W, Krawczyk L, Ładziński P, Grabarek BO, Jałowiecki P. Variations in Values of State, Response Entropy and Haemodynamic Parameters Associated with Development of Different Epileptiform Patterns during Volatile Induction of General Anaesthesia with Two Different Anaesthetic Regimens Using Sevoflurane in Comparison with Intravenous Induct: A Comparative Study. Brain Sci. 2020 Jun 12;10(6):366. doi: 10.3390/brainsci10060366. PMID: 32545600; PMCID: PMC7349226.

In the methodology section you writte: 

Normocarbia (35–45 
mmHg) was maintained -I would explain why - because hyperventilation may induce the presence of epileptiform patterns in the EEGs 

Vakkuri A, Jantti V, Särkelä M, Lindgren L, Korttila K, Yli-Hankala A. Epileptiform EEG during sevoflurane mask induction: effect of delaying the onset of hyperventilation. Acta Anaesthesiol Scand. 2000 Jul;44(6):713-9. doi: 10.1034/j.1399-6576.2000.440609.x. PMID: 10903015. 

what produces erroneus values of depth of anaesthesia monitors. 

In the limitation section I would welcome some more comments: 

  1. time delay between the detection of EEG patterns and their transformation into a digital signal may impair the actual depth of anaesthesia, which must be taken into consideration, what has not yet been studied with Phase lag entropy. Kreuzer M, Zanner R, Pilge S, Paprotny S, Kochs EF, Schneider G. Time delay of monitors of the hypnotic component of anesthesia: analysis of state entropy and index of consciousness. Anesth Analg. 2012 Aug;115(2):315-9. doi: 10.1213/ANE.0b013e31825801ea. Epub 2012 May 14. PMID: 22584557.
  2. the use of rocuronium was identified to result in an erroneous decrease of BIS values, and its influence on the Phase lag entropy value is unknown and requiries simillar studies. 
  3. presence of epileptiform patterns in patients EEGs were found to abnormally increase or decrease the state entropy values so and the influence on their possible presence in the present study on the Phase lag entropy value is unknown and requiries simillar further studies. Stasiowski M, Duława A, Szumera I, Marciniak R, Niewiadomska E, Kaspera W, Krawczyk L, Ładziński P, Grabarek BO, Jałowiecki P. Variations in Values of State, Response Entropy and Haemodynamic Parameters Associated with Development of Different Epileptiform Patterns during Volatile Induction of General Anaesthesia with Two Different Anaesthetic Regimens Using Sevoflurane in Comparison with Intravenous Induct: A Comparative Study. Brain Sci. 2020 Jun 12;10(6):366. doi: 10.3390/brainsci10060366. PMID: 32545600; PMCID: PMC7349226

Good luck for authors for a decent study, that I will gladly cite. :)

Author Response

We would like to thank you for taking the time to review our article. We have made some corrections and clarification in the manuscript after going over the reviewer's comment. Please see the attachment. We hope that the revised manuscript will better meet the requirements of your journal for publication.

This manuscript is a resubmission of an earlier submission. The following is a list of the peer review reports and author responses from that submission.